# A Mathematical Model for Predicting the Sauter Mean Diameter of Liquid-Medium Ultrasonic Atomizing Nozzle Based on Orthogonal Design

**Shilin Li [1], Gaogao Wu [1], Pengfei Wang [1,2,*], Yan Cui [1], Chang Tian [1] and Han Han [1]**

[1] School of Resource, Environment & Safety Engineering, Hunan University of Science & Technology, Xiangtan 411201, China; lishilin_1130@126.com (S.L.); wugaogao163@163.com (G.W.); cuiyan911@163.com (Y.C.); 13789307108@163.com (C.T.); hnkdhang@163.com (H.H.)

[2] Work Safety Key Lab on Prevention and Control of Gas and Roof Disasters for Southern Coal Mines, Hunan University of Science & Technology, Xiangtan 411201, China

\* Correspondence: pfwang@sina.cn

**Abstract:** As a new type of atomizing nozzle with superior atomizing performance, the liquid-medium ultrasonic atomization nozzle has been widely applied in the field of spray dust reduction. In this study, in order to establish a mathematical model for predicting the Sauter mean diameter (SMD) of such nozzles, the interaction between the SMD of the nozzle and the three influencing factors, i.e., air pressure, water pressure, and outlet diameter were investigated based on the custom-designed spraying experiment platform and orthogonal design methods. Through range analysis, it was obtained that the three parameters affecting the SMD of the nozzle are in the order of air pressure > water pressure > outlet diameter. On this basis, using the multivariate nonlinear regression method, the mathematical model for predicting the SMD of the nozzle was constructed. Comparison of the experimental results with the predicted values of the SMD of the nozzle by the multivariate nonlinear regression mathematical model, showed strong similarity with an average relative error of only about 5%. Therefore, the established mathematical model in this paper can be used to predict and calculate the droplet size for liquid-medium ultrasonic atomizing nozzles.

**Keywords:** spray dust reduction; liquid-medium ultrasonic atomizing nozzle; droplet size; orthogonal design; mathematical model



## 1. Introduction

A large amount of dust is produced from mining, metal smelting and other production activities [1–5]. The health and safety of workers who are exposed to high dust concentrations for long periods of time are at a serious risk [6–9]. According to the regulation from the National Health Committee of China, at production sites, the 8-h weighted average concentrations of total dust and respirable dust should be lower than 4.0 mg/m$^3$ and 2.5 mg/m$^3$, respectively. Meeting this regulation is a challenge for most production companies [10–14]. Nowadays, various production enterprises have adopted several effective dust protection measures; however, the situation of occupational disease in China is still severe, and pneumoconiosis remains the most serious occupational disease. As of the end of 2020, China had reported more than 1,012,000 cases of occupational diseases, and pneumoconiosis accounted for almost 88.6% of the total number of cases [15–17].

Spraying is one of the most commonly used dust management technologies in industrial fields [18,19]. In spraying-based dust reduction, the pressure nozzles are generally used to achieve water atomization [20–22]. However, due to the limited water supply pressure, the atomization effect of the pressure nozzles is often unsatisfactory, and the dust removal efficiency is generally low. Moreover, the traditional pressure nozzle consumes a large amount of water, which causes a certain degree of pollution in the industrial workplace [23–26]. Ultrasonic atomization is a method of using ultrasonic waves to break up

liquid into fine droplets. Compared with conventional pressure atomization, the advantages of ultrasonic atomization include low water pressure requirements, small amount of water consumption, and high dust removal efficiency (especially for respiratory dust). According to different working principles and internal structures, ultrasonic atomizing nozzles can be classified into two categories: piezoelectric atomizing nozzle and liquid-medium atomizing nozzle. Liquid-medium ultrasonic atomizing nozzles are mostly utilized in industrial dust removal sites, which use a resonant cavity to convert the high-speed fluid kinetic energy at the nozzle outlet into mechanical energy with wavy vibration, thereby generating ultrasonic waves [27,28].

In 1927, the ultrasonic atomization phenomenon was first discovered and preliminarily analyzed by Loomis and Woods (1927) [29]. In the middle of the 20th century, many scholars studied the dependence of the droplet size on the vibration frequency, and then a function of the droplet size on the vibration frequency was established. In addition, they pointed out that the surface tension wave was the primary cause for the formation of fine particles during the atomization process. In order to validate the relationship between the droplet size and the surface tension wave, Lang (1962) worked out the formulas between the median particle size and relevant factors such as the surface tension in the low frequency working environment through experiments [30]. Sarohia et al. (1979) proposed three modes of the resonant cavity and provided a clear classification and summary of the resonant mechanism of the resonant cavity, which further improved the resonant mechanism of the liquid-medium ultrasonic nozzle [31]. Based on the above three modes, Sobieraj et al. (1991) analyzed the vibration of the resonant cavity and the influence of the resonant mode by studying the nozzle outlet and the structure of the resonant cavity [32]. In the 21st century, the numerical simulation technology has been developed rapidly, and the combination of numerical simulation and experiments has been widely used. Hamed et al. (2003), Narayanan et al. (2009), and Kim et al. (2018) have successively performed numerical simulations on the unsteady flow in the resonant cavity of the liquid-medium ultrasonic nozzle. The results further enriched the research on the influencing factors for the working mode of the resonant cavity and determined the generation of ultrasonic location [33–35].

China's research on liquid-medium ultrasonic nozzles started late, but many scholars have performed extensive experimental and numerical simulation studies. Zhang et al. (2007, 2010) analyzed the principle and characteristics of liquid-medium ultrasonic atomization and examined the variation law of atomization quality with operating parameters and structural parameters [36]. Sun (2004) experimentally investigated the variation of acoustic parameters with the structure parameters of nozzles and established relevant empirical formulas [37]. Zhang et al. (2002) analyzed the ultrasonic atomization performance of water through orthogonal experiments, and the results showed that the number of droplets with the size smaller than 50 μm can be used as an index for the optimal parameter of the nozzle [38]. Using this index, the amount of water became the main influencing factor. In addition, a mathematical model for ultrasonic atomization performance was established through regression analysis of experimental data. Meanwhile, some scholars also used the fluid dynamics software ANYSY FLUENT to simulate the internal flow field, external flow field, and atomization performance of the fluid-medium ultrasonic atomizing nozzle, and to analyze the pressure and velocity vector distribution of the droplet field [39,40]. Li et al. (2017, 2018) and Gao et al. (2017) designed an ultrasonic atomizing nozzle with the new structure, analyzed the influence of the structure parameters of the resonant cavity and operating parameters of the nozzle on the internal flow field, and atomizing performance of the ultrasonic nozzle, and calculated the relationship between the cavity structure and sound pressure [41–43].

In summary, in the previous studies, scholars have focused on the flow field mode in the resonant cavity and ultrasonic frequency, and thoroughly studied the atomization mechanism of the liquid-medium ultrasonic nozzle. Meanwhile, some scholars have investigated the droplet size of this type of nozzle, and obtained simple prediction mathematical models for the droplet size. However, in the existing mathematical models, the influencing factors

for the droplet size were not completely considered, accordingly, the established droplet prediction model cannot be applied to the program development in engineering sites. In order to fill the gaps in existing research, this study analyzed the changes of the SMD of the nozzle with three parameters, i.e., air pressure, water pressure and outlet diameter using the orthogonal experimental design method. A Malvern real-time high-speed spray particle size analyzer, an intelligent electromagnetic flow meter, an air mass flow meter, and a high-performance camera were utilized in the study. Then, a mathematical model for predicting the SMD of the nozzle was constructed using the multivariate nonlinear regression analysis method, which can prove to be an effective tool for the parameter prediction for ultrasonic atomizing nozzle.

## 2. Experimental System and Scheme

### 2.1. Experimental System

In this study, the BL-CSBPZ-SS liquid-medium ultrasonic atomizing nozzle produced by Jining Bolin Spraying Equipment Co., Ltd., Jining China was used. The droplet flow was in a solid cone shape. The ultrasonic atomizing nozzle was mainly composed of a water inlet, an air inlet, a spray outlet, a mixing chamber, and a resonance cavity. Both the air inlet and water inlet had the inner diameter of 12.0 mm, and the specification of the connection was 1/2-inch internal thread. The exit diameter was in the range of 0.8~2.0 mm, as shown in Figure 1. The water inlet was located at the bottom of the nozzle, while the air inlet was arranged on the side of the nozzle, and the resonance cavity (ultrasonic generator) was located at the front of the nozzle outlet.

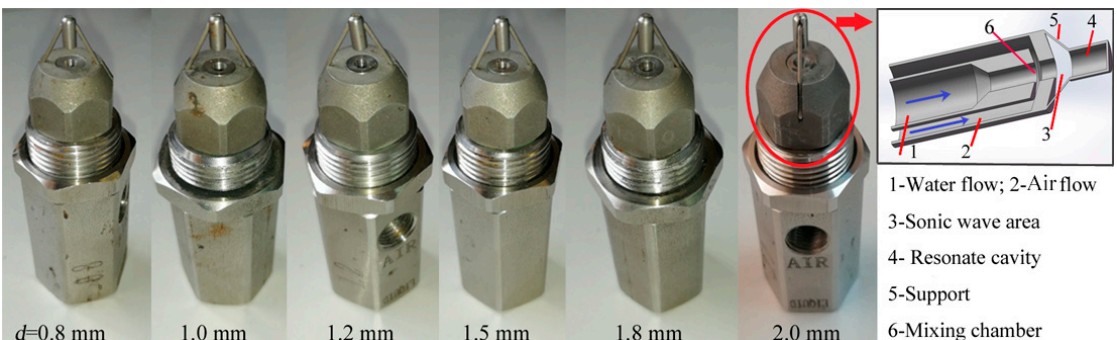

**Figure 1.** The ultrasonic atomizing nozzles used in experiment.

As shown in Figure 2, the experimental platform for the ultrasonic atomization nozzle consisted of a pump, a water tank, a control cabinet, an air compressor, a Malvern real-time high-speed spray particle size analyzer, a frequency converter, an intelligent electromagnetic flow meter, an air mass flow meter, a digital manometer, and some related pipes and valves. The output water pressure of the pump was adjusted by a frequency converter. The water supply pressure was measured by a digital manometer. The water flow rate was measured by an intelligent electromagnetic flow meter. The air pressure was controlled by a pressure reducing valve. The air flow rate was measured by an air mass flow meter. The distribution of droplet size in the droplet field was monitored by a high-speed spray particle size analyzer in real time [44,45].

### 2.2. Experimental Scheme

In this study, the orthogonal design method was used to analyze the atomization performance of the ultrasonic atomization nozzle. The orthogonal experimental design uses the minimum number of tests to analyze multiple factors and levels and can achieve the equivalent results of several comprehensive experiments. Before the orthogonal experiment was conducted, the experimental scheme was determined through the orthogonal table [46].

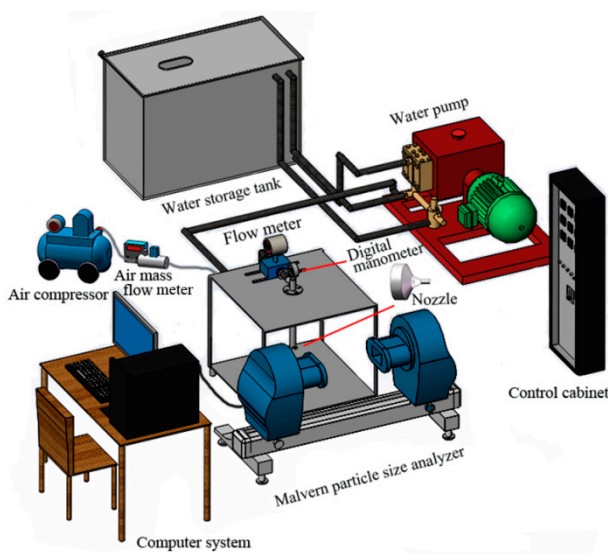

**Figure 2.** The experimental platform.

### 2.2.1. Factors and Levels of Orthogonal Experiment

The atomization characteristics of a nozzle typically consist of the flow rate and droplet size. The flow rate of liquid-medium ultrasonic atomizing nozzles includes both the air flow rate and water flow rate. Since the diameter of droplet particles was not uniformly distributed, the average diameter of the droplet group was used to represent the droplet size. At present, there are various methods for calculating the average particle diameter of droplets. The mass median diameter (MMD) and Sauter mean diameter (SMD) were the most commonly used parameters for the droplet size. In this study, SMD was used as an evaluation index for the fineness of droplets.

In the spraying-based dust reduction system at the engineering sites, both the atomizing medium and the external environment were relatively constant. For certain types of fluid-medium ultrasonic atomizing nozzles, the influencing factors of the atomizing parameters include air pressure ($p_{air}$), water pressure ($p_L$), and outlet diameter ($d$). Therefore, air pressure, water pressure, and outlet diameter were taken as the three factors of orthogonal design. Based on previous field surveys and actual measurement, the level ranges of these three factors were determined by comprehensive considering of the dust reduction efficiency, water consumption, and site conditions. An orthogonal experiment table was designed using the three factors and the determined levels of the factors. At the same time, in order to highlight the uniformity of the selected values and the gap of the results, the experimental scheme was designed with a fixed interval. Each factor was set to five levels, and the "three factors, five levels" L25 ($5^3$) orthogonal design method was used in the study. The factors and their levels in the orthogonal experiment are shown in Table 1.

**Table 1.** Factors and levels in the orthogonal experimental.

| NO. of Levels | Influence Factor | | |
|---|---|---|---|
| | $p_{air}$/MPa | $p_L$/MPa | $d$/mm |
| 1 | 0.2 | 0.2 | 0.8 |
| 2 | 0.3 | 0.3 | 1.0 |
| 3 | 0.4 | 0.4 | 1.2 |
| 4 | 0.5 | 0.5 | 1.5 |
| 5 | 0.6 | 0.6 | 1.8 |

### 2.2.2. Experimental Method for Atomization Parameters

According to the orthogonal experiment design scheme shown in Table 1, the atomization characteristics of the ultrasonic atomization nozzle were examined, including the

water flow ($Q_L$), air flow ($Q_{air}$), and droplet size. The air flow rate was measured by a D07-60B mass flow controller, and the water flow rate was measured by a YY-LED15K4C electromagnetic flow meter. The droplet size was measured by a Malvern droplet size analyzer. The Malvern droplet size analyzer is based on the line measurement principle, and therefore, the droplet data we obtained were the distribution of droplet particle size along the laser beam. In this experiment, the data were selected from the droplets located 50 cm in front of the nozzle outlet.

## 3. Experimental Results and Analysis

The influence of related factors on the atomization performance of the nozzle was experimentally studied according to the orthogonal design scheme in Table 1, and the results are shown in Table 2. Figure 3 shows the distribution of droplet size under 25 operating conditions, in which the red curve represents the cumulative percentage of the droplet size, and the blue column represents the volumetric frequency of the droplet size.

**Table 2.** Orthogonal experimental results.

| NO. | Influence Factor | | | Experiment Result | | |
|---|---|---|---|---|---|---|
| | $p_{air}$/MPa | $p_L$/MPa | $d$/mm | $Q_L$/(L·min$^{-1}$) | $Q_{air}$/(L·min$^{-1}$) | SMD/μm |
| 1 | 0.2 | 0.2 | 0.8 | 0.83 | 31 | 52.28 |
| 2 | 0.2 | 0.3 | 1.0 | 1.50 | 23 | 68.45 |
| 3 | 0.2 | 0.4 | 1.2 | 2.33 | 32 | 78.23 |
| 4 | 0.2 | 0.5 | 1.5 | 3.00 | 33 | 88.24 |
| 5 | 0.2 | 0.6 | 1.8 | 4.67 | 17 | 95.80 |
| 6 | 0.3 | 0.2 | 1.0 | 1.00 | 40 | 43.92 |
| 7 | 0.3 | 0.3 | 1.2 | 2.00 | 48 | 66.04 |
| 8 | 0.3 | 0.4 | 1.5 | 2.50 | 43 | 82.46 |
| 9 | 0.3 | 0.5 | 1.8 | 3.33 | 27 | 87.35 |
| 10 | 0.3 | 0.6 | 0.8 | 1.17 | 43 | 61.89 |
| 11 | 0.4 | 0.2 | 1.2 | 1.33 | 76 | 38.52 |
| 12 | 0.4 | 0.3 | 1.5 | 1.83 | 66 | 50.80 |
| 13 | 0.4 | 0.4 | 1.8 | 3.00 | 43 | 77.08 |
| 14 | 0.4 | 0.5 | 0.8 | 1.17 | 56 | 48.19 |
| 15 | 0.4 | 0.6 | 1.0 | 1.50 | 45 | 69.79 |
| 16 | 0.5 | 0.2 | 1.5 | 1.17 | 98 | 39.55 |
| 17 | 0.5 | 0.3 | 1.8 | 1.33 | 80 | 57.31 |
| 18 | 0.5 | 0.4 | 0.8 | 0.83 | 73 | 36.60 |
| 19 | 0.5 | 0.5 | 1.0 | 1.17 | 66 | 49.94 |
| 20 | 0.5 | 0.6 | 1.2 | 1.83 | 78 | 63.96 |
| 21 | 0.6 | 0.2 | 1.8 | 0.50 | 157 | 40.48 |
| 22 | 0.6 | 0.3 | 0.8 | 0.50 | 91 | 20.21 |
| 23 | 0.6 | 0.4 | 1.0 | 1.17 | 94 | 39.57 |
| 24 | 0.6 | 0.5 | 1.2 | 1.83 | 96 | 50.89 |
| 25 | 0.6 | 0.6 | 1.5 | 2.17 | 88 | 70.08 |

The orthogonal experimental results in Table 2 and Figure 3 show the wide distribution of the nozzle's atomization characteristic parameters, which can basically meet the requirements of investigation and analysis. The wide distribution of the experimental results indicated that the level of factors was not limited to local areas but can accurately reflect the overall situation of the factors, suggesting that the orthogonal experiment design scheme was reasonable and effective [47,48]. At industrial production sites, according to the requirements on the nozzle atomization parameters, a parameter combination that approximated the requirements of the conditions can be selected from the orthogonal experimental results in Table 2.

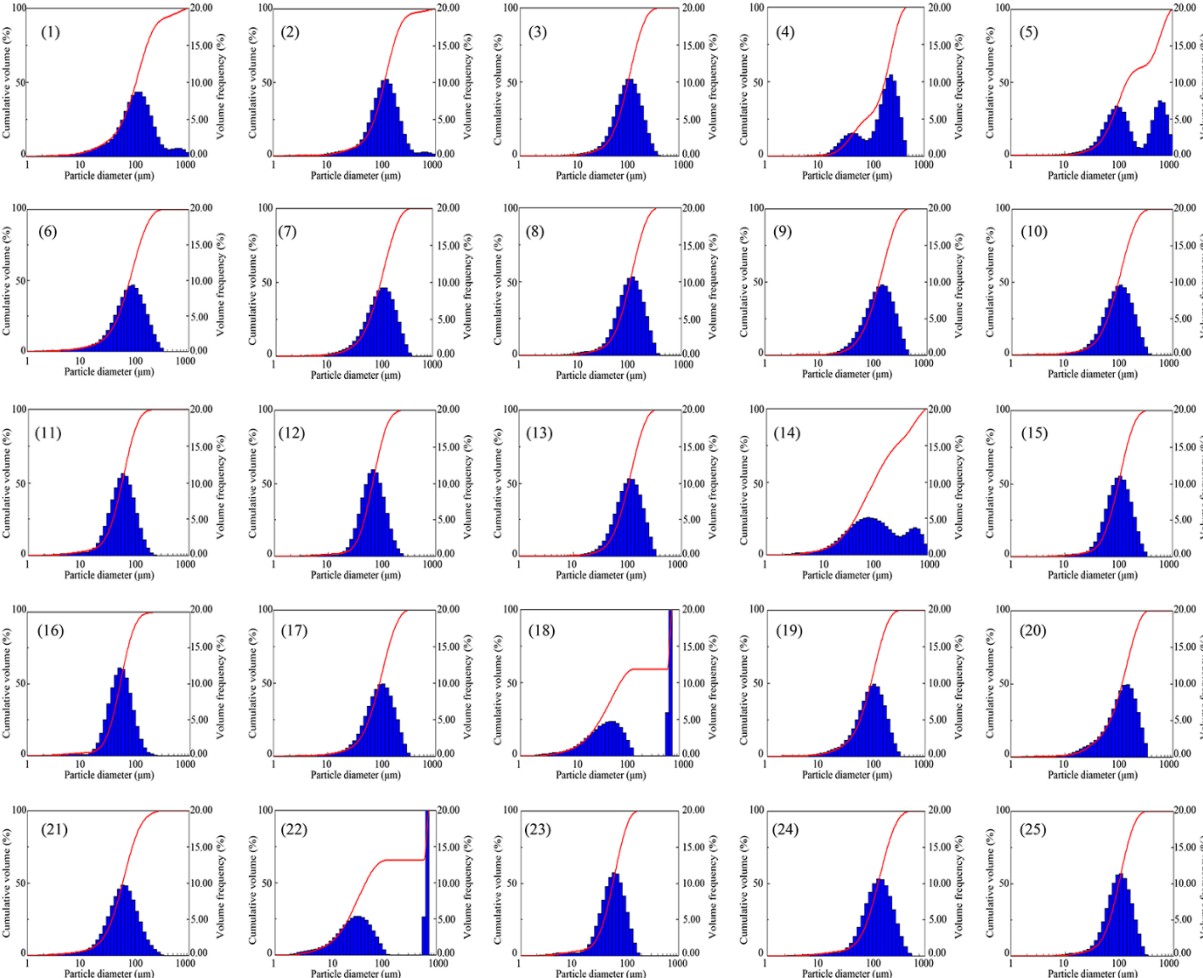

**Figure 3.** The droplet size distribution under 25 operating conditions.

By comprehensively averaging the SMD data in Table 2, the influencing significance of various factors on the SMD can be obtained through range analysis. Figure 4 shows the average value and range of SMD. From Figure 4, among the three influencing factors for the droplet size, the range of air pressure is the largest. The influencing significance of various factors can be ranked as $R_{air} > R_L > R_d$. From the overall trend of SMD, the SMD was negatively correlated with air pressure, positively correlated with water pressure and outlet diameter. The compressed air and low-speed water flow collided in the mixing chamber to achieve the primary atomization (initial atomization). Then, the gas–liquid two-phase flow was sprayed from the nozzle outlet at a high speed. The impulse of the fluid excited the ultrasonic generator to generate ultrasonic waves, which led to the secondary atomization of water.

The experimental results on the flow rate showed that at a higher air pressure, the water flow rate of the nozzle was lower and the air flow rate was greater. As the air flow rate increased, the velocity of the gas–liquid two-phase flow at the outlet increased, the intensity of the ultrasonic waves increased, and the generated effect on the surface of the droplets became more significant. As a result, the oscillation frequency of the bubbles was larger, and the particle size of droplets decreased. The increase of the outlet diameter led to a continuous increase in water flow rate. In contrast, at a relatively stable air flow rate, the atomization energy was reduced for a unit mass of water, which affected the effect of the primary and secondary atomization of the liquid. Consequently, the droplet got larger and larger. The increase in water pressure led to a greater water flow rate and a lower air flow rate. As a result, the air–liquid flow ratio continued to decrease, as shown in Figure 5. At a

lower air–liquid flow ratio of nozzle, the required atomization energy for a unit mass of droplets was decreased, thus both the primary atomization and the secondary atomization effects became weaker, and the droplet size increased.

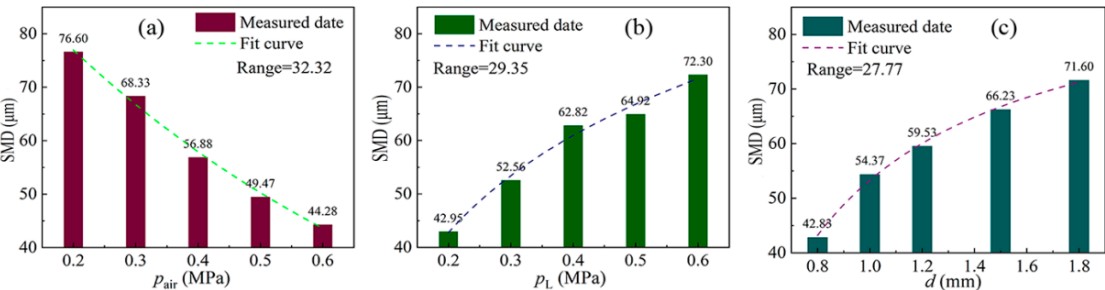

**Figure 4.** The relationship between SMD and three factors: (**a**) $p_{air}$; (**b**) $p_L$; and (**c**) $d$.

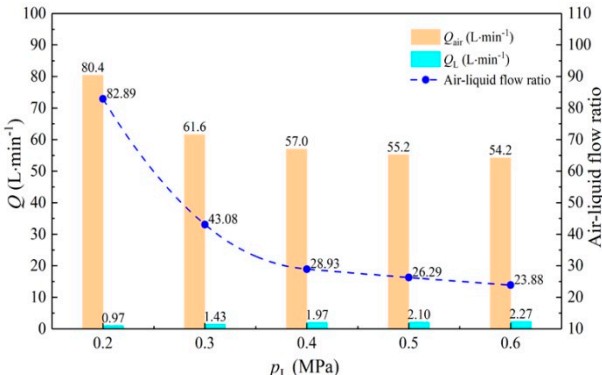

**Figure 5.** The impact of water pressure on the air–liquid flow ratio and the nozzle flow.

## 4. Establishment and Verification of Mathematical Model

In the previous section, the nozzle atomization parameters under 25 operating conditions were analyzed based on the orthogonal experiment design. However, it is difficult to study the relationship between the SMD of the nozzle and the influencing factors using traditional methods, because the influencing factors are diverse and complicated. Multivariate nonlinear analysis methods can greatly approximate measured data, thus constructing mathematical models that can realistically reveal the relationship between input and output variables. In this study, a multivariate nonlinear analysis method was adopted to establish the model for predicting the SMD of the nozzle.

### 4.1. Establishment of Mathematical Model

In the multivariate nonlinear regression method, the function form should be known first, and then the model coefficient can be fitted. To establish the mathematical model for predicting the SMD of the nozzle, several single-factor fitting formulas were firstly obtained, allowing for the best fitting formula to be determined through variance analysis and regression analysis. Next, the single-factor fitting formulas were synthesized into the multivariate nonlinear mathematical model. According to the change law of the air pressure and SMD, eight functions, including the linear function, logarithmic function, and S-shape function etc. were used in the SPSS software for fitting. The fitting results of each function are shown in Table 3. In this table, $R^2$ indicates the goodness of fit of the model, and the closer of $R^2$ to 1 is fit the better. $F$ is a statistic to test the significance of the formula, which is the ratio of the mean regression sum of squares to the mean residual sum of squares. The larger the $F$ is, the better the fit. The closer the regression significance is to 0, the better the fit [49].

**Table 3.** Fitting results of air pressure and SMD.

| Function | $R^2$ | ANOVA | | | | Parameters Estimated by Function | | | |
|---|---|---|---|---|---|---|---|---|---|
| | | $F$ | Degree of Freedom 1 | Degree of Freedom 2 | Regression Significance | Constant | Factor 1 | Factor 2 | Factor 3 |
| Linear | 0.984 | 181.650 | 1 | 3 | 0.001 | 92.512 | −83.500 | | |
| Logarithmic | 0.987 | 219.342 | 1 | 3 | 0.001 | 29.037 | −30.480 | | |
| S-shape | 0.903 | 27.794 | 1 | 3 | 0.013 | 3.593 | 0.161 | | |
| Exponential | 0.997 | 439.050 | 1 | 3 | 0.000 | 102.187 | −1.419 | | |
| Inverse function | 0.936 | 44.014 | 1 | 3 | 0.007 | 31.055 | 9.675 | | |
| Power function | 0.973 | 108.389 | 1 | 3 | 0.003 | 34.950 | −0.512 | | |
| Quadratic | 0.994 | 172.564 | 2 | 2 | 0.006 | 102.712 | −141.786 | 72.857 | |
| Cubic | 0.998 | 202.028 | 3 | 1 | 0.052 | 80.032 | 58.914 | −467.143 | 450.000 |

From Table 3, among the eight functions, the values of $R^2$ of the cubic function, exponential function and quadratic function are all greater than 0.99, indicating the better goodness of fit. The regression significance results of the model indicated that the exponential function shows the best fitting, with the maximum $F$ and the minimum regression significance. Meanwhile, according to the significance of regression coefficients in Table 4, the significance of the quadratic function and cubic function is far greater than that of the exponential function. Based on the above analysis, it was concluded that the variation law of the air pressure and SMD was in accordance with the exponential function.

**Table 4.** Significance of regression coefficients t of the function.

| Function | Fitting Formula | Significance of Regression Coefficient | | | |
|---|---|---|---|---|---|
| | | Constant | Coefficient 1 | Coefficient 2 | Coefficient 3 |
| Linear | $y = 92.512 − 83.5x$ | 0.000 | 0.001 | | |
| Logarithmic | $y = 29.037 − 30.48\ln(x)$ | 0.001 | 0.001 | | |
| S-shape | $y = e^{3.593+0.161/x}$ | 0.000 | 0.013 | | |
| Exponential | $y = 102.187e^{-1.419x}$ | 0.000 | 0.000 | | |
| Inverse function | $y = 31.055 + 9.675/x$ | 0.007 | 0.007 | | |
| Power function | $y = 34.95x^{-0.512}$ | 0.000 | 0.001 | | |
| Quadratic | $y = 102.712 − 141.786x + 72.857x^2$ | 0.003 | 0.044 | 0.197 | |
| Cubic | $y = 80.032 + 58.914x − 467.143x^2 + 450x^3$ | 0.118 | 0.727 | 0.403 | 0.359 |

For the change relationship between the $p_L$ and $d$ and SMD, the same analysis method was used to obtain the optimal fitting formula, and the optimal fitting formula between the three factors and SMD was summarized in Table 5.

**Table 5.** Fitting formula between single factor and SMD.

| Factor | Function Expression | Fitting Formula | $R^2$ |
|---|---|---|---|
| $p_{air}$ | $y_1 = b_1 e^{b_2 x_1}$ | $y_1 = 102.187e^{-1.419x_1}$ | 0.993 |
| $p_L$ | $y_2 = b_1 + b_2 \ln(x_2)$ | $y_2 = 84.978 + 26.216\ln(x_2)$ | 0.984 |
| $d$ | $y_3 = b_1 + b_2/x_3$ | $y_3 = 93.65 − 40.341/x_3$ | 0.996 |

According to the fitting formula between the single factor and SMD in Table 5, a multivariate nonlinear regression mathematical model was established [50,51]:

$$y = b_1 + b_2 e^{-1.419x_1} + b_3 \ln(x_2) + b_4/x_3 \tag{1}$$

where, $y$ represents the SMD of the droplet, μm; $x_1$ represents the air pressure ($p_{air}$), MPa; $x_2$ represents the water pressure ($p_L$), MPa; $x_3$ represents the outlet diameter ($d$), mm; $b_k$ represents the regression coefficient, $k = 1\sim4$. According to the measured 25 groups of experimental data, the multivariate nonlinear regression was selected in the SPSS software

and the model (1) was input to fit the model coefficient, and then the multivariate nonlinear regression mathematical model can be obtained as follows:

$$y = 54.799 + 106.504e^{-1.419x_1} + 25.198\ln(x_2) - 38.033/x_3 \tag{2}$$

In the multiple nonlinear regression Equation (2), $R^2 = 0.962$, indicating the better goodness of fit of the regression model. In order to further improve the goodness of fit and the accuracy of predicted values of the regression model, considering the interactions between various factors, the correction terms were added to the original model. Then, the modified multivariate nonlinear regression mathematical model can be expressed as:

$$y = b_1 + b_2e^{-1.419x_1} + b_3\ln(x_2) + b_4/x_3 + b_5x_1x_2 + b_6x_1x_3 + b_7x_2x_3 \tag{3}$$

According to the 25 groups of experimental data in Table 2, the multivariate nonlinear regression was selected again in the SPSS software to modify the prediction model. Then, with Equation (3) we can obtain the modified multivariate nonlinear regression mathematical model as follows:

$$\begin{aligned} y = -14.465 + 172.912e^{-1.419x_1} + 20.03\ln(x_2) - 30.721/x_3 \\ + 68.262x_1x_2 + 22.905x_1x_3 - 5.414x_2x_3 \end{aligned} \tag{4}$$

The value of $R^2$ in the modified prediction model was equal to 0.970, indicating that the goodness of fit of the multiple nonlinear regression was improved. The SMD of the nozzle can be calculated according to Equation (4).

### 4.2. Verification of Mathematical Model

The established model was used to calculate the SMD under 25 experimental conditions to verify the accuracy of the prediction model. Then, the calculation results were compared with the orthogonal experimental data, as shown in Figure 6. From the figure, it can be seen that the calculated value of SMD based on the prediction model was consistent with the orthogonal experimental results, with an average relative error of only 4.39%.

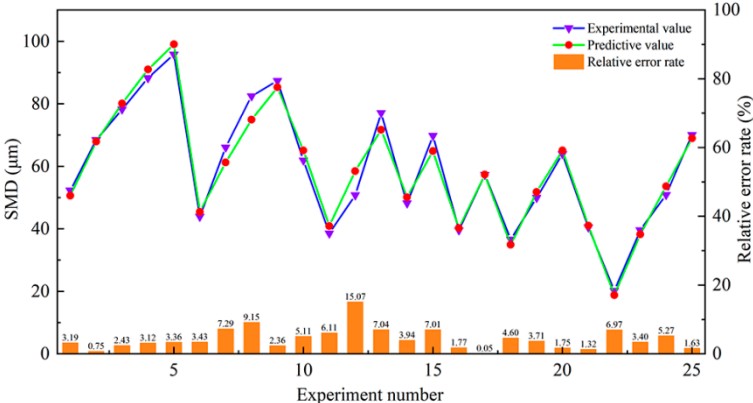

**Figure 6.** Comparison between the predicted values and orthogonal experimental values results of SMD.

In order to further verify the validity of the established prediction model, a nozzle was selected for experiment. In the experiment, the differences between the predicted SMD values of the nozzle and the experimental values were compared under different air pressure and water pressure. Figure 7 shows the change curve of the predicted SMD value and the experimental value of the nozzle with $d = 2.0$ mm. It can be seen from the figure that the predicted value has a consistent change trend with the experimental value, indicating that the model can qualitatively reflect the change rule of droplet particle size with the operating parameters. At the same time, the average relative error between the predicted value and the experimental value is only 4.89%. Considering the influence of the

experimental environment on the spray field as well as the complex environment of actual engineering site, there are many environmental factors affecting the particle size of droplets, thus, the model error is within the acceptable range. Therefore, it can be concluded that the prediction model established in this study has high accuracy and can be used for theoretical prediction and calculation of the SMD of nozzles.

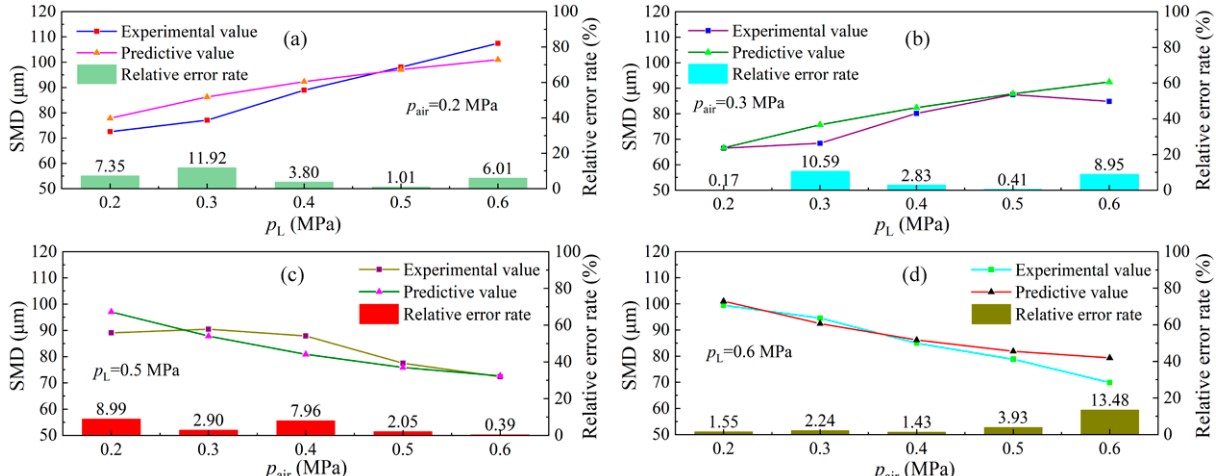

**Figure 7.** The predicted and experimental values of SMD for different air-water pressure nozzles: (**a**) $p_{air}$ = 0.2 MPa; (**b**) $p_{air}$ = 0.3 MPa; (**c**) $p_L$ = 0.5 MPa and (**d**) $p_L$ = 0.6 MPa.

The droplet size is an important parameter affecting the spray dust reduction. In practical application, the influencing factors of droplet size include air pressure, water pressure and outlet diameter. When the values of the three factors are known, the SMD of the nozzle can be predicted according to the prediction model. At the same time, the three factors can be changed according to the prediction model to obtain the appropriate droplet size. Due to the complexity and variability of the environment in practical application, arbitrary changes of the three factors are limited. Therefore, in order to obtain the appropriate droplet size, other factors can be adjusted through the prediction model under the condition that one factor is limited, as shown in Figure 8. It can be seen from the figure that the prediction model can be used to determine the nozzle droplet size and provide guidance for the actual application of nozzles. Overall, the variation law of the three factors and SMD plays an important role in the engineering application of nozzles, and the relationship between the three factors and SMD can be established through the prediction model.

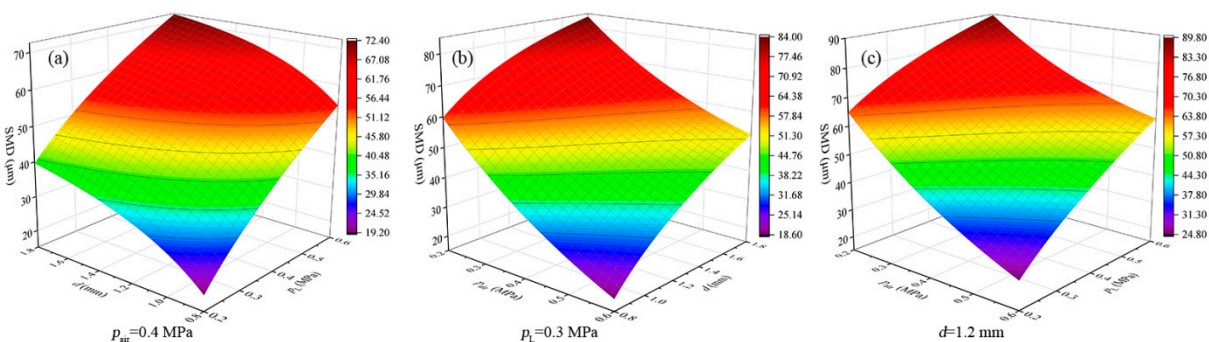

**Figure 8.** The relationship between the SMD of a single-factor constant nozzle and various factors: (**a**) $p_{air}$ = 0.4 MPa; (**b**) $p_L$ = 0.3 MPa; and (**c**) $d$ = 1.2 mm.

## 5. Conclusions

In this study, based on the orthogonal design method, the atomization parameters of a liquid-medium ultrasonic atomization nozzle were obtained under 25 operating conditions using a Malvern real-time high-speed spray particle size analyzer, an intelligent electromagnetic flow meter and an air mass flow meter. Then, the relationship between the atomization parameters and the influencing factors was analyzed by the means of average and range analysis. The atomization parameters in the analysis include the air flow rate, water flow rate, and droplet size, while the influencing factors include the air pressure, water pressure, and nozzle outlet diameter. On this basis, the multivariable nonlinear regression method was used to construct a model for predicting the SMD of nozzles. The predicted values of the SMD by the established mathematical model agreed with the experimental results, and the average relative error was only about 5%. The developed model can be used for theoretical prediction and calculation of the droplet size parameters of nozzles.

**Author Contributions:** Conceptualization, S.L. and P.W.; methodology, G.W.; software, G.W.; validation, G.W., C.T. and H.H.; formal analysis, G.W.; investigation, Y.C.; resources, P.W.; data curation, S.L.; writing—original draft preparation, S.L.; writing—review and editing, S.L.; visualization, G.W. All authors have read and agreed to the published version of the manuscript.

**Funding:** This research was funded by the Work Safety Key Lab on Prevention and Control of Gas and Roof Disasters for Southern Coal of China grant number E21728. This research was funded by the Scientific Research Project of Hunan Province Office of Education and Natural science foundation of Hunan grant number 19B197 and 2020JJ5172.

**Informed Consent Statement:** Informed consent was obtained from all subjects.

**Acknowledgments:** Project E21728 Supported by the Work Safety Key Lab on Prevention and Control of Gas and Roof Disasters for Southern Coal of China, Project 19B197 Supported by the Scientific Research Project of Hunan Province Office of Education, and Project 2020JJ5172 Supported by the Natural science foundation of Hunan, are gratefully acknowledged.

**Conflicts of Interest:** The authors declare no conflict of interest.

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
