# Peer review of "A Mathematical Model for Predicting the Sauter Mean Diameter of Liquid-Medium Ultrasonic Atomizing Nozzle Based on Orthogonal Design"

_applsci, doi:10.3390/app112411628_

Round 1

Reviewer 1 Report

In the manuscript "A mathematical model for predicting the Sauter mean diameter
of liquid-medium ultrasonic atomizing nozzle based on orthogonal design" by
Shilin Li et al., in order to establish a mathematical model for predicting the Sauter mean
diameter (SMD) of the liquid-medium ultrasonic atomization nozzles (widely applied in the 
field of spray dust reduction), the interaction between the SMD of the nozzle and the three 
influencing factors, i.e., air pressure, water pressure, and outlet diameter was investigated based on
the custom-designed spraying experiment platform and orthogonal design methods. As the result,
a mathematical model for predicting the SMD of the nozzle was constructed and compared 
with the results of the performed experiment.

The obtained results are interesting and useful but  the following should be corrected:

1. The literature should be in alphabetical  order. If the first author is the same,
alphabetical order should be applied to the second, and so on.

2. The English should be improved.

1. Abstract, line 7: three parameters ... is --> three parameters ... are

2. P. 3, l. 7: changes the --> changes of the

3. P. 3, last paragraph, l 1-2: nozzle consisted --> nozzle was consisted

4. P. 4,Sect. 2.2 l. 5: "experiment scheme" should be "experimental scheme" or 
"the scheme of experiment". This should be corrected throughout the paper. For example
instead of "experiment results" should be "experimental results" or
"results of experiment"

5. P. 4, l. 8 from the end: considering the --> considering of the

6. p. 5, Sect. 3, l. 2: experimented studied --> experimentally studied

7. p. 8, l. 1: is inductive the better ?!?

8. p. 8, l. 3, below Table 3: mdel --> model

9. p. 8, l. 1 below Table 4: For the change relationship between the other ?!?

Author Response

Thanks to the experts for their careful review of the paper. Based on your revision comments, I have carefully revised the paper. The references in the paper have all been revised to be sorted alphabetically. All the places where the translation of the paper is not in place have been revised according to your suggestions.

Reviewer 2 Report

The article presents a mathematical model for predicting SMD liquid-medium ultrasonic atomizing nozzle. Overall the article is well written, follows a logical structure, has a good explanation of used methods and obtained results.  

Few minor mistakes can be found, such as in Fig.1. the "2-Air flow" is written incorrectly as "2-Ari flow". In Table 1 water pressure symbol is capital, while in the text it is not capital. The document does not contain a nomenclature list, therefore the symbols are described in the text. While it is known, that Q is used for flow rate and units support that, it would be good to explain the meaning of Q in Table 2. Small misspelling mistakes have been marked in the attached file. 

The article is clearly a continuation of similar previous studies as Wang et al. 2019e,f,g. It would be beneficial for the scientific field if this continuation would be mentioned by authors and somehow conclusions or improvements between articles would be used to benefit the scientific field and development of the studies. 

In subsection 2.2.1 it is mentioned that "both the atomizing medium and the external environment were relatively constant". It is not clear what is meant by the external environment being relatively constant. Some explanations would be beneficial - are some specific sites being evaluated and what are these statements based on. Two other unclear uses of words are marked for review. 

 In the article nonlinear regression is used as a method to mathematically describe the nozzle. R-squared is used throughout to evaluate the best method. Since the use of R-squared in nonlinear regression is questionable, some comments on particular choices would be beneficial. 

In the article, there is no mention of the accuracy of the used experimental method, which limits the evaluation of verification results. This also creates a question on the choice of the number of significant figures used in the article for relative error. Display of uncertainties would help to see if mathematical model results are within the uncertainties. 

Figures 7 and 8 both contain multiple graphs - for example - (a), (b) and so on. But there is no mention of them in the article. At least the captions should describe the differences between the graphs. 

Author Response

Thanks to the experts for their careful review of the paper. Based on your revision comments, I have carefully revised the paper. All the places where the translation of the paper is not in place have been revised according to your suggestions. The introduction of the thesis mentioned that Wang's research is a continuation of this article. The relative stability of the external environment refers to the relative stability of the environment's temperature, humidity, and pressure. R2 is a parameter describing the goodness of fit. The paper experiment adopts orthogonal experiment, which is a scientific and mature experiment method. The captions of Figure 7 and Figure 8 have been modified.